# OpenReview forum: "SafeRBench: A Comprehensive Benchmark for Safety Assessment of Large Reasoning Models"
_ICLR.cc/2026/Conference — ICLR 2026 Conference Withdrawn Submission_

### Official Review · Reviewer_w8Yi · 2025-10-15

**Soundness:** 3
**Presentation:** 3
**Contribution:** 3
**Rating:** 4
**Confidence:** 2

**Summary:**

The paper introduces a benchmark SafeRBench that assesses large reasoning models safety from inputs, and intermediate reasoning to outputs. through structured input characterization, micro-thought chunking for fine-grained analysis, and human-aligned safety evaluation

**Strengths:**

Existing benchmarks have many limitations. SafeRBench categorize queries
by risk levels, accounting for affected groups and severity of impact, and construct a balanced
benchmark dataset that reflects diverse harm gradients. And introduced trace evaluation to do fine-grained analysis of risk propagation.

**Weaknesses:**

LLMs and reasoning models evolve rapidly, so it’s unclear whether the benchmark’s scope will remain broad or generalizable for emerging models. The paper also lacks theoretical and mechanistic insight—it reads more like a collection and categorization of GPT-generated data. While it provides useful diagnostic metrics such as scores and rankings, it does not explain why models fail or exhibit unsafe behaviors.

**Questions:**

1. Could you elaborate on the motivation and intuition behind the different risk categories, and explain how these risks are defined and categorized?
2. The **segmentation of reasoning traces** appears to be a crucial step, yet it remains unclear how the **BERT-based** and **LLM-based** methods are applied in this process. Since segmentation can significantly influence the final outcomes, could you clarify how the **granularity of segmentation** is determined?
3. Is the **segmentation** applied **only to long text inputs**, or does it also affect shorter reasoning traces?

---

### Official Review · Reviewer_G1bD · 2025-10-23

**Soundness:** 2
**Presentation:** 3
**Contribution:** 2
**Rating:** 2
**Confidence:** 4

**Summary:**

This paper proposes SafeRBench, a benchmark and framework for evaluating the safety of large reasoning models (LRMs). On the input level, risk is categorized into 6 classes with 3 possible risk levels. Outputs are evaluated based on refusal, risk level (4 levels), and execution level. Reasoning traces are evaluated by risk density, defense density, intent awareness, and safe strategy conversion. Using this evaluation framework, the authors evaluated 19 LRMs and compared them to draw insights on how different model and reasoning features impact safety.

**Strengths:**

This work evaluates the safety of reasoning traces, which haven't been thoroughly evaluated before. This work proposes comprehensive metrics to quantify the safety of LRMs, providing more fine grained, multi dimensional understanding of how and why LRMs are unsafe. The paper is clearly written.

**Weaknesses:**

I have some concerns about the robustness of some of the proposed metrics and evaluation. I think the manuscript in its current state does not make a sufficiently substantial contribution to be accepted by ICLR without major revision and additional experimentation.

1. One of the main claims of novelty is that this work evaluates reasoning trace safety. However, based on this paper, it's unclear why it's insufficient to evaluate the input + output without reasoning traces. It would strengthen the paper if the authors could provide quantification of how much reasoning traces matter. For example, use an LLM judge to evaluate how safe (input + output) is vs. (input + reasoning trace + output).

2. Some of the evaluation metrics need justification and clarification. See my questions in the section below

3. lines 52-53: "Existing benchmarks mainly annotate the risk category of outputs, such as Safety-Bench (Zhang et al., 2024b) and HarmBench (Mazeika et al., 2024)" - I believe HarmBench only provides inputs, so the risk categories should be on inputs. SafetyBench seems to be the same. Many existing benchmarks label risk categories on inputs and not outputs, such as WildGuard, AIR-Bench, etc. This contradicts the claim that input risk categories is a novelty of the current work.

4. fig. 2: it's a stretch to call a low-medium-high scale a spectrum, when it only adds an intermediate level to binary categorization. nvidia/Aegis-AI-Content-Safety-Dataset-1.0 contains labels categorized into safe, needs caution, and unsafe (with specific risk category). this is similar to the stratified risk levels of the present work, which undermines the claimed novelty

5. fig. 3: please find alternative ways to illustrate the evaluation results. Currently, it's very hard to recognize which line corresponds to which model

**Questions:**

1. In lines 42-43, can you explain what "incremental capability scaffolding, rationale laundering, or late-stage revelation" are more clearly and how these vulnerabilities might arise (e.g., are they emergent or subject to external attacks)?
2. Line 53: "This limits their effectiveness for LRMs, where long reasoning traces introduce layered risks" - why and how?
3. Line 155-158: Some risk categories inherently target groups rather than individuals (e.g., environmental & global threats, social safety). are distributions of risk levels balanced for each risk category?
4. Line 160: How were the queries generated?
5. Why is Response Complexity a safety metric? It doesn't seem to distinguish models well based on Figure 3.
6. Lines 252-258: Are there any patterns in which the risk score evolve over micro chunks in the reasoning traces? e.g., plot s over t.
7. Why does trajectory coherence measure safety?
8. Fig. 4: some metrics seem highly correlated - do we need all of them or can they be condensed into more independent metrics?

---

### Official Review · Reviewer_3Wq5 · 2025-11-01

**Soundness:** 3
**Presentation:** 2
**Contribution:** 2
**Rating:** 2
**Confidence:** 4

**Summary:**

The paper proposes a new benchmark and multiple new metrics to evaluate reasoning models.
Specifically, it proposes a taxonomy for creating a set of evaluation prompts. The evaluation then performs a chunking approach on the reasoning chains and evaluates each reasoning chunk. They also compare human annotations with AI annotations. The benchmark evaluates safety across 10 dimensions, including intent awareness and risk level.

**Strengths:**

The strengths of the paper include the following:

- interesting conclusions: The finding that for small models the thinking setup increases risk, while for medium ones it does not, and for bigger ones the risk is increased again, is very interesting. I also found the discussion on stronger tail controls insightful.
- Chunking: the use of chunking is interesting and could have been expanded upon more.
- The paper provides a thorough correlation analysis.

**Weaknesses:**

The main weaknesses of the paper are the following:

- Novelty: There exists a lot of published papers that provide a finer-grained analysis of risk levels of the input already, such as: Li, Jing-Jing, et al. "Safetyanalyst: Interpretable, transparent, and steerable safety moderation for ai behavior.", or Zhang, Yuyou, et al. "Safety is Not Only About Refusal: Reasoning-Enhanced Fine-tuning for Interpretable LLM Safety.". It would be good to get a better understanding of what makes this paper here different and the benchmark more suitable than other approaches.
- Clarity: The paper could improve the clarity of writing, see questions below.
- Chunking: It seems like the chunking ended up being done by GPT-5 via prompting. In lines 184-186 it says that this is because GPT-5 is better than previous chunking models, but the paper does not provide any numbers to support that claim. How did you evaluate it? Also, does it mean one needs to pay for GPT to evaluate ones models’ on the benchmark? Wouldn’t it be more efficient to host a smaller model for that purpose? In general, I am a bit concerned that it requires so much prompting of proprietary models to run this benchmark.

**Questions:**

- How did you come up with the taxonomy?
- 184-186: did you actually evaluate this? Do you have some numbers for that?
- Fig 3: very hard to read
- 248: Confusing metric: why are words per sentence a good indicator for density?
- 252: why does the chunk index matter?
- 369: does ability to infer user intent also prevent over-refusal?
- 431: would be good to have an example, as it’s very hard to understand what a high risk query is

---

### Official Review · Reviewer_iFhf · 2025-11-03

**Soundness:** 3
**Presentation:** 3
**Contribution:** 3
**Rating:** 6
**Confidence:** 5

**Summary:**

SafeRBench introduces the first comprehensive benchmark designed to evaluate the safety of Large Reasoning Models (LRMs) throughout their full reasoning process, spanning inputs, intermediate reasoning traces, and final outputs. Unlike prior LLM safety benchmarks that focus only on surface-level harms, SafeRBench captures process-level risks unique to LRMs, such as harmful rationales or late-stage toxic reasoning. It provides a three-layer evaluation framework: (1) a stratified dataset of 1,128 harmful queries categorized across six risk domains and three risk levels; (2) micro-thought chunking that segments reasoning traces into fine-grained cognitive intents for detailed risk analysis; and (3) ten safety dimensions grouped into Risk Exposure and Safety Awareness scores. Experiments on 19 LRMs show that reasoning traces strongly predict safety outcomes: models with higher Intention Awareness and Defense Density produce safer responses. Medium-sized “thinking” models perform best, while very large models exhibit an “always-help” bias that can reintroduce risk. SafeRBench thus establishes a scalable, human-aligned framework for diagnosing and improving LRM safety across reasoning dynamics

**Strengths:**

- SafeRBench uniquely assesses model safety across the entire reasoning pipeline including from input prompts to intermediate reasoning traces to final outputs, capturing risks that traditional output-only benchmarks miss.
- By analyzing reasoning traces, SafeRBench detects latent and evolving safety failures, such as rationale laundering or late-stage harmful reasoning.
- It introduces a compact yet representative dataset of 1,128 harmful queries, systematically balanced across six harm categories and three risk tiers, enabling precise and reproducible evaluations.
- The introduced micro-thought chunking mechanism is thoughtful, which segments long reasoning traces into semantically coherent units labeled with cognitive intents, allowing fine-grained, interpretable analysis of reasoning safety.
- SafeRBench aligns LLM-based safety judgments with human annotations.
- Model evaluation is comprehensive and insightful.

Overall this work provides useful work for advancing reasoning model safety.

**Weaknesses:**

- The fine-grained segmentation and evaluation of long reasoning traces may be computationally expensive, restricting scalability to larger datasets or more models without significant resources. It'll be good to conduct some cost analysis to show the computation overhead.
- The benchmark's results hinges on the choice of evaluator models.

**Questions:**

See Weaknesses + How does different evaluator models may change the benchmarking results?

---

### Note · Authors · 2025-11-26

**Comment:**

We would like to express our sincere gratitude for the time and effort you have dedicated to reviewing our paper. We truly appreciate your recognition of the strengths of our work, as well as your constructive criticism regarding its limitations.

After carefully considering the feedback, we have decided to withdraw our manuscript at this stage to address the issues raised and further improve the quality of our work.

I have read and agree with the venue's withdrawal policy on behalf of myself and my co-authors.

**Withdrawal Confirmation:**

I have read and agree with the venue's withdrawal policy on behalf of myself and my co-authors.